# Attention over learned object embeddings enables complex visual reasoning

**David Ding**     **Felix Hill**     **Adam Santoro**     **Malcolm Reynolds**     **Matt Botvinick**

DeepMind
London, United Kingdom
{fding, felixhill, adamsantoro, mareynolds, botvinick}@google.com

## Abstract

Neural networks have achieved success in a wide array of perceptual tasks but often fail at tasks involving both perception and higher-level reasoning. On these more challenging tasks, bespoke approaches (such as modular symbolic components, independent dynamics models or semantic parsers) targeted towards that specific type of task have typically performed better. The downside to these targeted approaches, however, is that they can be more brittle than general-purpose neural networks, requiring significant modification or even redesign according to the particular task at hand. Here, we propose a more general neural-network-based approach to dynamic visual reasoning problems that obtains state-of-the-art performance on three different domains, in each case outperforming bespoke modular approaches tailored specifically to the task. Our method relies on learned object-centric representations, self-attention and self-supervised dynamics learning, and all three elements together are required for strong performance to emerge. The success of this combination suggests that there may be no need to trade off flexibility for performance on problems involving spatio-temporal or causal-style reasoning. With the right soft biases and learning objectives in a neural network we may be able to attain the best of both worlds.

## 1   Introduction

Despite the popularity of artificial neural networks, a body of recent work has focused on their limitations as models of cognition and reasoning. Experiments with dynamical reasoning datasets such as CLEVRER [41], CATER [12], and ACRE [44] show that neural networks can fail to adequately reason about the spatio-temporal, compositional or causal structure of visual scenes. On CLEVRER, where models must answer questions about the dynamics of colliding objects, previous experiments show that neural networks can adequately *describe* the video, but fail when asked to *predict*, *explain*, or consider *counterfactual* possibilities. Similarly, on CATER, an object-tracking task, models have trouble tracking the movement of objects when they are hidden in a container. Finally, on ACRE, a dataset testing for causal inference, popular models only learned correlations between visual scenes and not the deeper causal logic.

Failures such as these on reasoning (rather than perception) problems have motivated the adoption of pipeline-style approaches that combine a general purpose neural network (such as a convolutional block) with a task-specific module that builds in the core logic of the task. For example, on CLEVRER the NS-DR method [41] applies a hand-coded symbolic logic engine (that has the core logic of CLEVRER built-in) to the outputs of a "perceptual" neural front-end, achieving better results than neural network baselines, particularly on counterfactual and explanatory problems. One

---

Model Code: https://github.com/deepmind/deepmind-research/tree/master/object_attention_for_reasoning.

35th Conference on Neural Information Processing Systems (NeurIPS 2021).

limitation of these pipeline approaches, however, is that they are typically created with a single problem or problem domain in mind, and may not apply out-of-the-box to other related problems. For example, to apply NS-DR to CATER, the entire symbolic module needs to be rewritten to handle the new interactions and task logic of CATER: the custom logic to handle collisions and object removal must be replaced with new custom logic to handle occlusions and grid-resolution, and these changes require further modifications to the perceptual front-end to output data in a new format. This brittleness is not exclusive to symbolic approaches. While Hungarian-matching between object embeddings may be well-suited for object-tracking tasks [45], it is not obvious how it would help for causal inference tasks.

Here, we describe a more general neural-network-based approach to visual spatio-temporal reasoning problems, which does not rely on task-specific integration of modular components. In place of these components, our model relies on three key aspects:

- Self-attention to effectively integrate information over time
- Soft-discretization of the input at the most informative level of abstraction – above pixels and local features, and below entire frames—corresponding approximately to 'objects'
- Self-supervised learning, i.e. requiring the model to infer masked out objects, to extract more information about dynamics from each sample.

While many past models have applied each individual ingredient separately (including on the tasks we study), we show that it is the *combination of all three ingredients in the right way* that allows our model to succeed.

The resulting model, which we call *Aloe* (Attention over Learned Object Embeddings), outperforms both pipeline and neural-network-based approaches on three different task domains designed to test physical and dynamical reasoning from pixel inputs. We highlight our key results here:

- **CLEVRER** (explanatory, predictive, and counterfactual reasoning): *Aloe* achieves significantly higher accuracy than both more task-specific, modular approaches, and previous neural network methods on all question types. On counterfactual questions, thought to be most challenging for neural-only architectures, we achieve **75%** vs **46%** accuracy for more specialised methods.
- **CATER** (object-permanence): *Aloe* achieves accuracy exceeding or matching other current models. Notably, the strongest alternative models were expressly designed for object-tracking, whereas our architecture is applicable without modification to other reasoning tasks as well.
- **ACRE** (causal-inference "beyond the simple strategy of inducing causal relationships by covariation" [44]): Overall, *Aloe* achieves **94%** vs the **67%** accuracy achieved by the top neuro-symbolic model. On the most challenging tasks, we achieve, for "backward-blocking" inference, **94.48%** (vs **16.06%** by the best modular, neuro-symbolic systems), and, for "screen-off" inference, **98.97%** (vs **0.00%** by a CNN-BERT baseline).

As we have emphasized, the previous best performing models for each task all contain task-specific design elements, whereas *Aloe* can be applied *to all the tasks without modification*. On CLEVRER, we also show that *Aloe* matches the performance of the previous best models with 40% less training data, which demonstrates that our approach is data-efficient as well as performant.

## 2    Methods

A guiding motivation for the design of *Aloe* is the converging evidence for the value of self-attention mechanisms operating on a finite sequences of discrete entities. Written language is inherently discrete and hence is well-suited to self-attention-based approaches. In other domains, such as raw audio or vision, it is less clear how to leverage self-attention. We hypothesize that the application of self-attention-based models to visual tasks could benefit from an approximate 'discretization' process, and determining the right level of discretization is an important choice that can significantly affect model performance.

At the finest level, data could simply be discretized into pixels (as is already the case for most machine-processed visual data). Pixels are too fine-grained for many applications, however—for one, the memory required to support self-attention across all pixels is prohibitive. Partly for this reason,

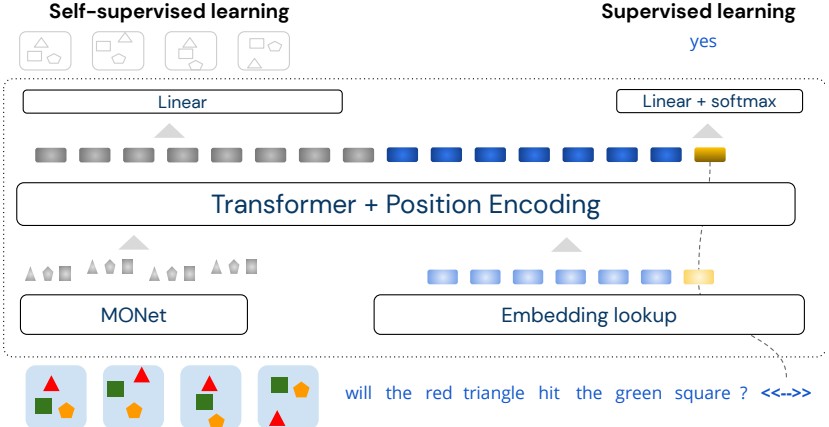

Figure 1: A schematic of the model architecture. See the main text for details.

coarser representations, such as the downsampled "hyper-pixel" outputs of a convolutional network, are often used instead (e.g. [27, 43]). In the case of videos, previous work considered even coarser discretization schemes, such as frame or subclip level representations [35].

The neuroscience literature, however, suggests that biological visual systems infer and exploit the existence of *objects*, rather than spatial or temporal blocks with artificial boundaries [5, 30, 32]. Because objects are the atomic units of physical interactions, it makes sense to discretize on the level of objects. Numerous object segmentation algorithms have been proposed [15, 19, 29]. We chose to use MONet, an unsupervised object segmentation algorithm [2]. Because MONet is unsupervised, we can train it directly in our domain of interest without the need for object segmentation labels. We emphasize that our choice of MONet is an implementation detail, and in Appendix B, we show that our framework of attention over learned object embeddings also works with other object-segmentation schemes. We also do not need to place strong demands on the object segmentation algorithm, e.g. for it to produce aligned output or to have a built-in dynamics model.

To segment each frame into object representations, **MONet** uses a recurrent attention network to obtain a set of $N_o$ "object attention masks" ($N_o$ is a fixed parameter). Each attention mask represents the probability that any given pixel belongs to that mask's object. The pixels assigned to the mask are encoded into latent variables with means $\mu_{ti} \in \mathbb{R}^d$, where $i$ indexes the object slot and $t$ the frame. These means are used as the object embeddings in *Aloe*. More details are provided in Appendix A.1.

The **self-attention component** is a transformer model [37] operating on a sequence of vectors in $\mathbb{R}^d$: the object representations $\mu_{ti}$ for all $t$ and $i$, a trainable vector $CLS \in \mathbb{R}^d$ used to generate classification results (analogous to the CLS token in BERT [9]), and (for CLEVRER) the embedded words $\mathbf{w}_i$ from the question (and choice for multiple choice questions). For the object representations $\mu_{ti}$ and word embeddings $\mathbf{w}_i$, we append a two-dimensional one-hot vector to $\mu_{ti}$ and $\mathbf{w}_i$ to indicate whether the input is a word or an object. Because the transformer is shared between the modalities, information can flow between objects and words to solve the task, as we show in Section 3.1.

We pass this sequence of vectors through a transformer with $N_T$ layers. All inputs are first projected (via a linear layer and ReLU activation) to $\mathbb{R}^{N_H \times d}$, where $N_H$ is the number of self-attention heads. We add a relative sinusoidal positional encoding at each layer of the transformer to give the model knowledge of the word and frame order [7]. The transformed value of $CLS$ is passed through an MLP (with one hidden layer of size $N_H$) to generate the final answer. A schema of our architecture is shown in Figure 1.

Note that in the model presented above (which we call *global attention*), the transformer sees no distinction between objects of different frames (other than through the position encoding). Another intuitive choice, which we call *hierarchical attention*, is to have one transformer acting on the objects of each frame independently, and another transformer acting on the concatenated outputs of the first transformer (this temporal division of input data is commonly used, e.g. in [35]). In pseudo-code, global attention can be expressed as

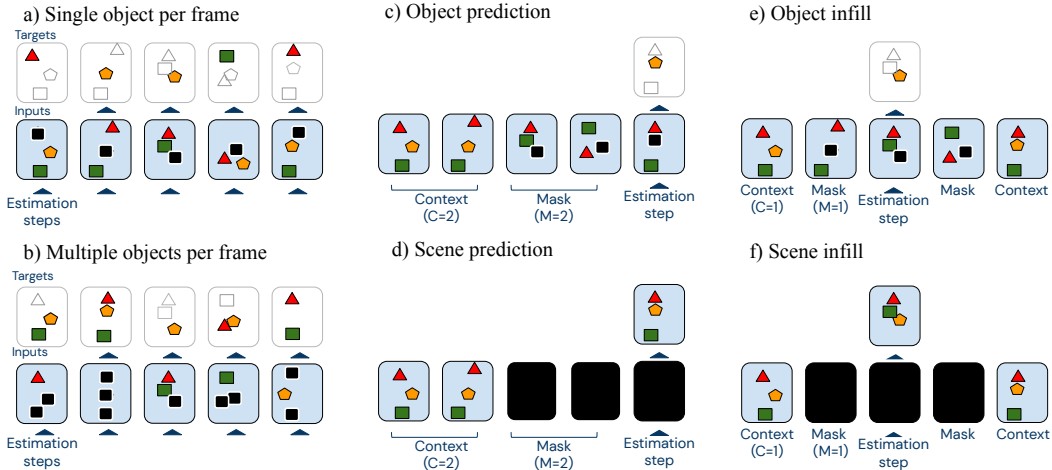

Figure 2: Different masking schemes for self-supervised learning applied to *Aloe*.

```
out = transformer(reshape(objects, [B, F * N, D])
```

and hiearchical attention as

```
out = transformer1(reshape(objects, [B * F, N, D]))

out = transformer2(reshape(out, [B, F, N * D])) .
```

We study the importance of global attention (objects as the atomic entities) vs hierarchical attention (objects, and subsequently frames as the atomic entities). The comparison is shown in Table 1.

## 2.1 Self-supervised learning

We explored whether self-supervised learning could improve the performance of *Aloe* beyond the benefits conveyed by object-level representation, i.e. in ways that support the model's interpretation of scene dynamics rather than just via improved perception of static observations. Our approach is inspired by the loss used in BERT [9], where a transformer model is trained to predict certain words that are masked from the input. In our case, we mask *object embeddings*, and train the model to infer the content of the masked object representations using its knowledge of unmasked objects.

Concretely, during training, we multiply each MONet latent $\mu_{ti}$ by a masking indicator, $m_{ti} \in \{0, 1\}$. Let $\mu'_{ti}$ be the transformed value of $m_{ti}\mu_{ti}$ after passing through the transformer. We expect the transformer to understand the underlying dynamics of the video, so that the masked out slot $\mu_{ti}$ could be predicted from $\mu'_{ti}$. To guide the transformer in learning effective representations capable of this type of dynamics prediction, we add an auxiliary loss:

$$\text{auxiliary loss} = \sum_{t,i} \tau_{ti} l\left(f(\mu'_{ti}), \mu\right),$$

where $f$ is a learned linear mapping to $\mathbb{R}^d$, $l$ a loss function, and $\tau_{ti} \in \{0, 1\}$ are one-hot indicator variables identifying the prediction targets (not necessarily just the masked out entries, since the prediction targets could be a subset of the masked out entries). We propagate gradients only to the parameters of $f$ and the transformer and not to the learned word and $CLS$ embeddings. This auxiliary loss is added to the main classification loss with weighting $\lambda$, and both losses are minimized simultaneously by the optimizer. We do not pretrain the model with only the auxiliary loss.

We tested two different loss functions for $l$, an L2 loss and a contrastive loss (formulas given in Appendix A.2), and six different masking schemes (settings of $m_{ti}$ and $\tau_{ti}$), as illustrated in Figure 2. This exploration was motivated by the observation that video inputs at adjacent timesteps are highly correlated in a way that adjacent words are not. We thus hypothesized that BERT-style prediction of adjacent words might not be optimal. A different masking strategy, in which prediction targets are

separated from the context by more than a single timestep, may stimulate capacity in the network to acquire knowledge that permits context-based unrolls and better long-horizon predictions.

The simplest approach would be to set $m_{ti} = 1$ uniformly at random across $t$ and $i$, fixing the expected proportion of the $m_{ti}$ set to 1 (schema $b$ in Figure 2). The targets would simply be the unmasked slots, $\tau_{ti} = 1 - m_{ti}$. One potential problem with this approach is that multiple objects could be masked out in a single frame. MONet can unpredictably switch object-to-slot assignments multiple times in a single video. If multiple slots are masked out, the transformer cannot determine with certainty which missing object to assign to each slot. Thus, the auxiliary loss could penalize the model even if it predicted all the objects correctly. To avoid this problem, we also try constraining the mask such that exactly one slot is masked out per frame (schema $a$).

To pose harder prediction challenges, we can add a buffer between the context (where $m_{ti} = 1$) and the infilling targets (where $\tau_{ti} = 1$). For $t$ in this buffer zone, both $m_{ti} = 0$ and $\tau_{ti} = 0$ (schemas $c$–$f$). We choose a single cutoff $T$ randomly, and we set $m_{ti} = 0$ for $t < T$ and $m_{ti} = 1$ for $t \geq T$. In the presence of this buffer, we compared prediction (where the context is strictly before the targets; schema $c, d$) versus infilling (where the context surrounds the targets; schema $e, f$). We also compared setting the targets as individual objects (schema $c, e$) versus targets as all objects in the scene (schema $d, f$). We visually inspect the efficacy of this self-supervised loss in encouraging better representations (beyond improvements of scores on tasks) in Appendix D.

# 3 Experiments

We tested *Aloe* on three datasets, CLEVRER [41], CATER [12], and ACRE [44]. For each dataset, we pretrained a MONet model on individual frames. More training details and a table of hyperparameters are given in Appendix A.3; these hyperparameters were obtained through a hyperparameter sweep. All error bars are standard deviations computed over at least 5 random seeds.

## 3.1 CLEVRER

CLEVRER features videos of CLEVR objects [21] that move and collide with each other. For each video, several questions are posed to test the model's understanding of the scene. Unlike most other visual question answering datasets, which test for only descriptive understanding ("what happened?"), CLEVRER poses other more complex questions, including explanatory questions ("why did something happen?"), predictive questions ("what will happen next?"), and counterfactual questions ("what would happen in a unseen circumstance?") [41].

We compare *Aloe* to state-of-the-art models reported in the literature: MAC (V+) and NS-DR [41], as well as the DCL model [6] (simultaneous to our work). MAC (V+) (based on the MAC network [20]) is an end-to-end network augmented with object information and trained using ground truth labels for object segmentation masks and features (e.g. color, shape). NS-DR and DCL are hybrid models that apply a symbolic logic engine to outputs of various neural networks. The neural networks are used to detect objects, predict dynamics, and parse the question into a program, and the symbolic executor runs the parsed program to obtain the final output. NS-DR is trained using ground truth labels and ground truth parsed programs, while DCL requires only the ground truth parsed programs.

Table 1 shows the result of *Aloe* compared to these models. Across all categories, *Aloe* significantly outperforms the previous best models. Moreover, compared to the other models, *Aloe* does not use any labeled data other than the correct answer for the questions, nor does it require pretraining on any other dataset. *Aloe* also was not specifically designed for this task, and it straightforwardly generalizes to other tasks as well, such as CATER [12] and ACRE [44]. We provide a few sample model classifications on a randomly selected set of videos and questions in Appendix E.1 and detailed analysis of counterfactual questions in Appendix C. These examples suggest qualitatively that, for most instances where the model was incorrect, humans would plausibly furnish the same answer.

**Attention analysis** (More analyses are given in Appendix D) We analyzed the cross-modal attention between question-words and the MONet objects. For each word, we determined the object that attended to that word with highest weight (for one head in the last layer). In the visualization below, the bounding boxes show the objects found by MONet, and each word is colored according to the

| Model | Descriptive | Explanatory | Predictive | Counterfactual |
|---|---|---|---|---|
| MAC (V+) | 86.4 | 22.3 | 42.9 | 25.1 |
| NS-DR | 88.1 | 79.6 | 68.7 | 42.2 |
| DCL | 90.7 | 82.8 | 82.0 | 46.5 |
| *Aloe* | **94.0** $\pm$ 0.4 | **96.0** $\pm$ 0.6 | **87.5** $\pm$ 3.0 | **75.6** $\pm$ 3.8 |
| *Aloe* $-$ self-attention + MLP | 45.4 | 16.0 | 27.7 | 9.9 |
| *Aloe* $-$ object-repr. + ResNet | 74.9 | 66.1 | 58.3 | 32.4 |
| *Aloe* $-$ global + hierarchical attn. | 80.6 | 87.4 | 73.5 | 55.1 |
| *Aloe* $-$ self-supervised loss | 91.0 | 92.8 | 82.8 | 68.7 |

Table 1: Performance (per question accuracy) on CLEVRER of *Aloe* compared to results from literature and to ablations: 1) MLP instead of self-attention; 2) ResNet superpixels instead of MONet objects; 3) hierarchical frame-level and intra-frame attention instead of global cross-frame object attention; 4) no auxiliary loss.

object that attended to it with highest weight (black represents a MONet slot without any objects). We observe that generally, objects attend heavily to the words that describe them.

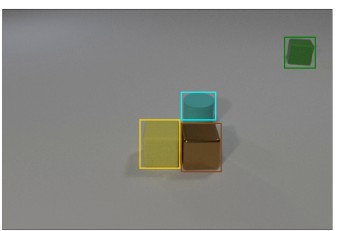

**Q:** If the cylinder is removed, which event will not happen?

1. The brown object collides with the green object.

2. The yellow object and the metal cube collide.

3. The yellow cube collides with the green object.

We also looked at the objects that were most heavily attended upon in determining the final answer. The image below illustrates the attention weights for the $CLS$ token attending on each object (for one head in the last layer), when the model is tasked with assessing the first choice of the question above. The bounding boxes show the two most heavily attended upon objects for one transformer head. We observe that this head focuses on the green and brown objects (asked about in choice 1), but switches its focus to the cyan cylinder when it looks like the cylinder might collide with the cubes and change the outcome.

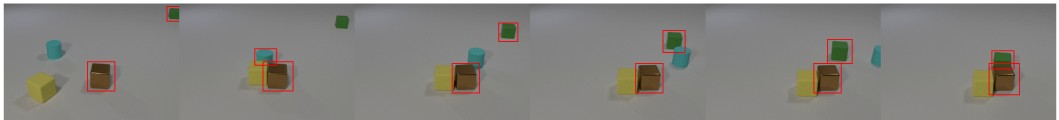

**Model ablation**   Table 1 shows the contributions of various components of *Aloe*. First, self-attention is necessary for solving this problem. For comparison, we replace *Aloe*'s transformer with four fully connected layers with 2048 units per layer[1]. We find that an MLP is unable to answer non-descriptive questions effectively, despite using more parameters (20M vs 15M parameters).

Second, we verify that an object-based discretization scheme is essential to the performance of *Aloe*. We compare with a version of the architecture where the MONet object representations $\mu_{ti}$ are replaced with ResNet hyperpixels as in Zambaldi et al. [43]. Concretely, we flatten the output of the final convolutional layer of the ResNet to obtain a sequence of feature vectors that is fed into the transformer as the discrete entities. To match MONet's pretraining regimen, we pretrain the ResNet on CLEVR [21] by training an *Aloe* model (using a ResNet instead of MONet) on the CLEVR task and initializing the ResNet used in the CLEVRER task with these pre-trained weights. We find that an object level representation, such as one output by MONet, greatly outperforms the locality-aware but object-agnostic ResNet representation.

We also observe the importance of global attention between all objects across all frames, compared to a hierarchical attention model where objects within a frame could attend to each other but frames could only attend to each other as an atomic entity. We hypothesize that global attention may be

---

[1]We also tried a bidirectional LSTM, which achieved even lower performance. This may be because the structure of our inputs requires the learning of long-range dependencies.

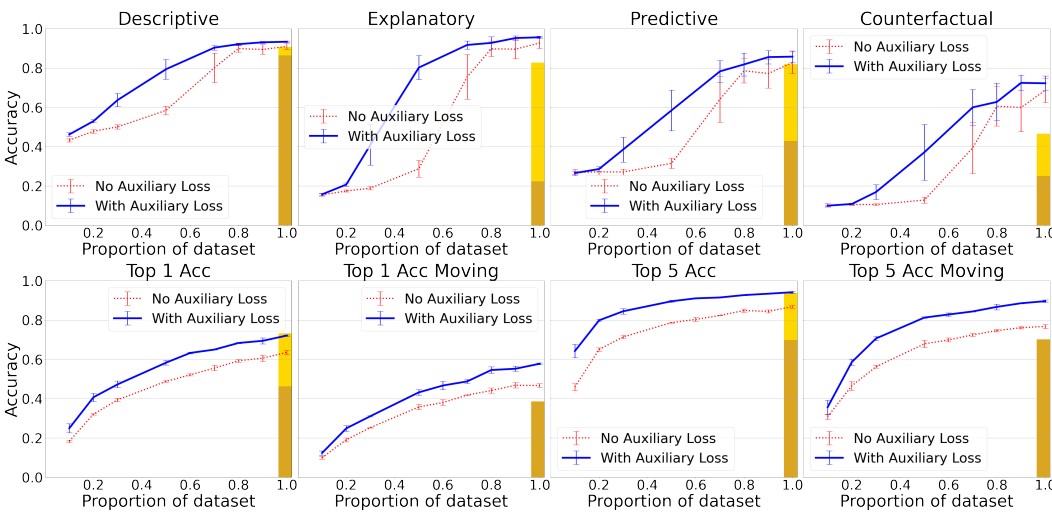

Figure 3: Accuracy with/without auxiliary loss for different proportions of CLEVRER (row 1) and CATER (row 2) training data. We also show comparisons with previous and concurrent work. For CLEVRER, the lighter yellow bar represents the best neurosymbolic model DCL, and the darker yellow bar represents the previous best distributed model, MAC (V+). For CATER, the lighter yellow bar represents Hopper and the darker yellow bar represents R3D+NL, the best published results for the moving camera dataset.

important because with hierarchical attention, objects in different frames can only attend to each other at the "frame" granularity. A cube attending to a cube in a different frame would then gather information about the other non-cube objects, muddling the resulting representation.

Finally, we see that an auxiliary self-supervised loss improves the performance of the model by between 4 and 6 percentage points, with the greatest improvement on the counterfactual questions.

**Self-supervision strategies**    We compared the various masking schemes and loss functions for our auxiliary loss; a detailed figure is provided in Appendix A (Figure 4). We find that for all question types in CLEVRER, an L2 loss performs better than a contrastive loss, and among the masking schemes, masking one object per frame is the most effective. This particular result runs counter to our hypothesis that predictions or infilling in which the target is temporally removed from the context could encourage the model to learn more about scene dynamics and object interactions than (BERT-style) local predictions of adjacent targets. Of course, there may be other settings or loss functions that reveal the benefits of non-local prediction or constrastive losses; we leave this investigation to future work.

**Data efficiency**    We investigated how model performance varies as a function of the number of labelled (question-answer) pairs it learns from. To do so, we train models on $N\%$ of the videos and their associated labeled data. We evaluate the effect of including the auxiliary self-supervised loss (applied to the entire dataset, not just the labelled portion) in this low data regime. This scenario, where unlabeled data is plentiful while labeled data is scarce, occurs frequently in practice, since collecting labeled data is much more expensive than collecting unlabeled data.

Figure 3 shows that our best model reaches the approximate level of the previous state-of-the-art approaches using only 50%-60% of the data. The self-supervised auxiliary loss makes a particular improvement to performance in low-data regimes. For instance, when trained on only 50% of the available labelled data, self-supervised learning enables the model to reach a performance of 37% on counterfactual questions (compared to 25% by MAC (V+) and 42% by NS-DR on the full dataset), while without self-supervision, the model only reaches a performance of 13% (compared to the 10% achieved by answering randomly [41]).

| Model | Top 1 (S) | Top 5 (S) | L1 (S) | Top 1 (M) | Top 5 (M) | L1 (M) |
|---|---|---|---|---|---|---|
| R3D LSTM | 60.2 | 81.8 | 1.2 | 28.6 | 63.3 | 1.7 |
| R3D + NL LSTM | 46.2 | 69.9 | 1.5 | 38.6 | 70.2 | 1.5 |
| OPNet | **74.8** | - | 0.54 | - | - | - |
| Hopper | 73.2 | 93.8 | 0.85 | - | - | - |
| *Aloe* (no auxiliary) | 60.5 | 84.5 | 0.90 | 46.8 | 75.1 | 1.3 |
| *Aloe* | 70.6 | 93.0 | 0.53 | 56.6 | 87.0 | 0.82 |
| *Aloe* (with L1 loss) | $74.0 \pm 0.3$ | $\mathbf{94.0 \pm 0.4}$ | $\mathbf{0.44 \pm 0.01}$ | $\mathbf{59.7 \pm 0.5}$ | $\mathbf{90.1 \pm 0.6}$ | $\mathbf{0.69 \pm 0.01}$ |

Table 2: Performance on CATER of *Aloe* compared to the best results from literature. We report top 1 accuracy, top 5 accuracy, and L1 distance between the predicted grid cell and true grid cell. The labels (S) and (M) refer to static and moving cameras.

## 3.2 CATER

In a second experiment, we tested *Aloe* on CATER, a widely-used object-tracking dataset [12, 14, 31, 45]. In CATER, objects from the CLEVR dataset [21] move and potentially occlude other objects, and the goal is to predict the location of a target object (called the *snitch*) in the final frame. Because the snitch could be occluded by multiple objects that could move in the meantime, a successful model must be sensitive to notions of object permanence. CATER also includes a moving camera variant, which introduces additional complexities for the model.

Concretely, CATER is setup as a classification challenge. Objects are located in an $xyz$ coordinate system, where x and y range from -3 to 3. The $xy$ plane is divided into a 6 by 6 grid, and the task is to predict the grid index of the snitch in the final frame. For *Aloe*, we use a classification loss (cross entropy over the 36 possible grid indices) and an L1 loss (L1 distance between predicted grid cell and the true grid cell).

Table 2 shows *Aloe* compared to state-of-the-art models in the literature on both static and moving camera videos. R3D and R3D NL are the strongest two models evaluated by Girdhar and Ramanan [12]. OPNet, or the Object Permanence Network [31], is an architecture with inductive biases designed for object tracking tasks; it was trained with extra supervised labels, namely the bounding boxes for all objects (including occluded ones). Hopper is a multi-hop transformer model developed simultaneously with this work [45]. One key component of Hopper is Hungarian matching between objects of different frames, a strong inductive bias for object tracking.

We train *Aloe* simultaneously on both static and moving camera videos. *Aloe* outperforms the R3D models for both static and moving cameras. We also ran *Aloe* with an additional auxiliary loss consisting of the L1 distance between the predicted cell and the actual cell. With this additional loss, we get comparable results in the *moving* camera case as the R3D models for the *static* camera case. Moreover, we achieve comparable accuracy as OPNet for accuracy and L1 distance, despite requiring less supervision to train. Appendix E.2 gives a few sample outputs from *Aloe*; in particular we note that it is able to find the target object in several cases where the object was occluded, demonstrating that *Aloe* is able to do some level of object tracking. Finally, we find that an auxiliary self-supervised loss helps the model perform well in the low data regime for CATER as well, as shown in Figure 3.

## 3.3 ACRE

Finally, we measured *Aloe*'s performance on ACRE, a causal induction dataset inspired by the Blicket task from developmental psychology [13, 44]. ACRE is divided into a set of problems. In each problem, certain objects are chosen to be "Blickets", and this assignment changes across problems. Each problem presents a context of six images to the model, where different objects are placed on a Blicket machine that lights up if one of those objects is a Blicket. The model is asked whether an unseen combination of objects will light up the Blicket machine. Besides "yes" and "no", a third possible answer is "undetermined", which is the case if it is impossible to determine for certain if the objects will light up the machine. Correct inference goes beyond mere correlation: even if every context scene involving object A has a lit-up machine, A's Blicketness is still uncertain if each of those scenes can potentially be explained by another object (deduction of A's Blicketness is *backward-blocked*).

| Model | All (C) | D.R. | I.D. | S.O. | B.B. | All (S) | D.R. | I.D. | S.O. | B.B |
|-------|---------|------|------|------|------|---------|------|------|------|-----|
| CNN-BERT | 43.79 | 54.07 | 46.88 | 40.57 | 28.79 | 39.93 | 55.97 | 68.25 | 0.00 | 45.59 |
| NS-OPT | 69.04 | 92.5 | 76.05 | 88.33 | 13.48 | 67.44 | 94.73 | **88.38** | 82.76 | 16.06 |
| *Aloe* | **91.76** | **97.14** | **90.8** | **96.8** | **78.81** | **93.90** | **97.18** | 71.24 | **98.97** | **94.48** |

Table 3: Performance on ACRE of *Aloe* compared to the best results from Zhang et al. [44], split across inference type (D.R=Direct, I.D=Indirect, S.O=Screen-Off, B.B=Backwards Blocking) and generalization type (C=Compositional, S=Systematic).

Inference problems in ACRE are categorized by reasoning type: reasoning from *direct* evidence (one of the context frames show the query objects on a machine), reasoning from *indirect evidence* (Blicketness must be deduced by combining evidence from several frames), *screened-off* reasoning (presence of non-Blickets do not matter if a single Blicket is present), and *backward-blocked* reasoning (Blicketness cannot be deduced due to confounding variables). Please see Zhang et al. [44] for a more detailed discussion of these reasoning types.

Table 3 show *Aloe* performance compared to a CNN-BERT baseline and to NS-OPT, a neuro-symbolic model introduced in Zhang et al. [44]. *Aloe* outperforms all extant models for almost all reasoning types and train-test splits. We did not need to do any tuning to apply our model to ACRE—settings from CATER yielded the reported results on the first attempt. Contrary to widely-held opinions that neural networks cannot generalize, *Aloe* generalizes in scenarios where the training and test sets contain different visual features (compositional split) or different numbers of activated machines in the context (systematic split). Moreover, *Aloe* achieved by far the best performance on the backward-blocking task, which requires the model to "go beyond the simple covariation strategy to discover the hidden causal relations" [44], dispelling the notion that neural networks can only find correlation. Comparison with NS-OPT (which uses object representations) and CNN-BERT (which uses attention) shows that neither object representations nor attention alone is sufficient for the task; combining these two ideas, as done in *Aloe* for instance, is essential for this complex reasoning task as well.

## 4   Related work

**Self-attention for reasoning**   Various studies have shown that transformers [37] can manipulate symbolic data in a manner traditionally associated with symbolic computation. For example, in Lample and Charton [23], a transformer model learned to do symbolic integration and solve ordinary differential equations symbolically, tasks traditionally reserved for symbolic computer algebra systems. Similarly, in Hahn et al. [17], a transformer model learned to solve formulas in propositional logic and demonstrated some degree of generalization to out of distribution formulas. Finally, Brown et al. [1] showed that a transformer trained for language modeling can also do simple analogical reasoning tasks without explicit training. Although these models do not necessarily beat carefully tuned symbolic algorithms in all cases (especially on out of distribution data), they are an important motivation for our proposed recipe for attaining strong reasoning capabilities from self-attention-based models on visually grounded tasks.

**Object representations**   A wide body of research points to the importance of object segmentation and representation learning (see e.g. Garnelo and Shanahan [11] for a discussion). Various methods have been proposed for object detection and feature extraction [2, 10, 15, 19, 25, 26, 29]. Past research have also investigated using object based representations in downstream tasks [8, 28].

**Self-supervised learning**   Another line of research concerns learning good representations through self-supervised learning, with an unsupervised auxiliary loss to encourage the discovery of better representations. These better representations could lead to improved performance on supervised tasks, especially when labeled data is scarce. In Devlin et al. [9], for instance, an auxiliary infill loss allows the BERT model to benefit from pretraining on a large corpus of unlabeled data. Our approach to object-centric self-supervised learning is heavily inspired by the BERT infilling loss. Other studies have shown similar benefits to auxiliary learning in vision as well [4, 16, 18]. These works apply various forms of contrastive losses to predict scene dynamics, and the better representations that result carry downstream benefits to supervised and reinforcement learning tasks.

**Vision and language in self-attention models**    Recently, many works have emerged on applying transformer models to visual and multimodal data, for static images [24, 27, 33, 36] and videos [34, 35, 43]. These approaches combine the output of convolutional networks with language in various ways using self-attention. While these previous works focused on popular visual question answering tasks, which typically consist of descriptive questions only [41], we focus on understanding deeper causal dynamics of videos. Together with these works, we provide more evidence that self-attention between visual and language elements enables good performance on a diverse set of tasks.

In addition, while the use of object representations for discretization in tasks involving static images is becoming more popular, the right way to discretize videos is less clear. We provide strong evidence in the form of ablation studies for architectural decisions that we claim are essential for higher reasoning for this type of data: visual elements should correspond to physical objects in the videos and inter-frame attention between sub-frame entities (as opposed to inter-frame attention of entire frames) is crucial. We also demonstrate the success of using unsupervised object segmentation methods as opposed to the supervised methods used in past work.

## 5    Conclusion

We have presented *Aloe*, a model that obtains state-of-the-art performance on three different task domains involving spatiotemporal reasoning about objects. In each of these tasks, previous state-of-the-art results were established by models with modular, task-specific components. *Aloe*, by contrast, is a unified solution to all three domains. Its flexibility comes from a reliance on only soft biases and learning objectives: self-attention over learned object embeddings and self-supervised learning of dynamics. We believe the simplicity of this approach is its strength, and hope that this fact, together with the provided code, makes it easy for others to adopt and apply to arbitrary spatio-temporal reasoning problems.

On many of these spatio-temporal reasoning problems, previous state-of-the-art was achieved by neuro-symbolic models [6, 11, 40, 41, 44]. Compared to neuro-symbolic models, *Aloe* can more easily be adapted to other tasks. Indeed, the symbolic components of neuro-symbolic models are often task-specific and not straightforwardly applicable to other tasks. Neuro-symbolic models do have a few advantages, however. First, they are often easier to interpret. Despite the insights that can be gleaned from *Aloe*'s attention weights, these soft computations are harder to interpret than the explicit symbolic computation found in neuro-symbolic models. Moreover, neuro-symbolic models can be structured in a more modular fashion, which can enable effective generalization to sub-tasks of the task on which the model was trained [6].

*Aloe* also has some important limitations. First, it has only been applied to synthetic datasets. This limitation is mainly due to the lack of real-world datasets that test for higher-order spatiotemporal reasoning, although we are excited that new datasets such as Traffic QA will be released soon [38]. Second, while the domains where *Aloe* is applied have been widely adopted and well-received by the research community, it remains possible that they do not evaluate the capacities that they aim to evaluate because of hidden biases or other factors. Regardless, we hope that this work stimulates the design and development of more challenging tasks that more closely approximate the ultimate goal of human or super-human-level visual, spatiotemporal and causal reasoning. Finally, from an ethical point of view, our model may share the common drawback of deep-learning models in perpetuating biases found in the training data, especially when applied to real world data. Development of causal reasoning models could also invite problematic applications involving automated assignment of blame.

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
