# A  Methods details

## A.1  MONet

To segment each $w \times h$ frame $F_t$ into $N_o$ object representations, MONet uses a recurrent attention network to obtain $N_o$ attention masks $\mathbf{A}_{ti} \in [0,1]^{w \times h}$ for $i = 1, \ldots, N_o$ that represent the probability of each pixel in $F_t$ belonging to the $i$-th object, with $\sum_{i=1}^{N_o} \mathbf{A}_{ti} = 1$. This attention network is coupled with a component VAE with latents $\mathbf{z}_{ti} \in \mathbb{R}^d$ for $i = 1, \ldots, N_o$ that reconstructs $\mathbf{A}_{ti} \odot F_t$, the $i$-th object in the image. The latent posterior distribution $q(\mathbf{z}_t | F_t, \mathbf{A}_{ti})$ is a diagonal Gaussian with mean $\mu_{ti}$, and we use $\mu_{ti}$ as the representation of the $i$-th object.

When these representations are fed into the transformer, we use a linear projection to map the raw object/word embeddings, which lie in $\mathbb{R}^d$, to a vector in $\mathbb{R}^{dN_H}$, where $N_H$ is the number of self-attention heads. This step is necessary as generally the latent dimensionality of MONet, $d$, is less than $N_H$ whereas a transformer expects the embedding size to be divisible by $N_H$.

## A.2  Self-supervised training

Recall in the main text that we wrote the auxiliary self-supervised loss as

$$\text{auxiliary loss} = \sum_{t,i} \tau_{ti} l\left(f(\mu'_{ti}), \mu\right).$$

We tested an L2 loss and a contrastive loss (inspired by the loss used in [18]), and the formulas for the two losses are respectively:

$$l_{\text{L2}}\left(f(\mu'_{ti}), \mu\right) = \|f(\mu'_{ti}) - \mu_{ti}\|_2^2$$

$$l_{\text{contrastive}}\left(f(\mu'_{ti}), \mu\right) = -\log \frac{\exp(f(\mu'_{ti}) \cdot \mu_{ti})}{\sum_{s,j} \exp\left(f(\mu'_{ti}) \cdot \mu_{sj}\right)}.$$

A comparison of these losses and the masking schemes is given in Figure 4.

We also tested a few variations of the contrastive loss inspired by literature and tested all combinations of variations. The first variation is where the negative examples all come from the same frame:

$$l_{\text{contrastive}}\left(f(\mu'_{ti}), \mu\right) = -\log \frac{\exp(f(\mu'_{ti}) \cdot \mu_{ti})}{\sum_{j} \exp\left(f(\mu'_{ti}) \cdot \mu_{tj}\right)}.$$

The second variation is adding a temperature $\tau$ to the softmax [4]:

$$l_{\text{contrastive}}\left(f(\mu'_{ti}), \mu\right) = -\log \frac{\exp(f(\mu'_{ti}) \cdot \mu_{ti})/\tau}{\sum_{s,j} \exp\left(f(\mu'_{ti}) \cdot \mu_{sj}/\tau\right)}.$$

The final variation we tested is using cosine similarity instead of dot product:

$$l_{\text{contrastive}}\left(f(\mu'_{ti}), \mu\right) = -\log \frac{\exp(\text{sim}(f(\mu'_{ti}), \mu_{ti}))}{\sum_{s,j} \exp\left(\text{sim}(f(\mu'_{ti}), \mu_{sj})\right)}.$$

where $\text{sim}(\mathbf{x}, \mathbf{y}) = \frac{\mathbf{x} \cdot \mathbf{y}}{\|\mathbf{x}\| \cdot \|\mathbf{y}\|}$. We found that these variations did not significantly change the performance of the model (and the optimal temperature setting was close to $\tau = 1$), and leave to future work more careful analysis of these contrastive losses and the representations they encourage.

## A.3  Training details

We generally follow similar training procedures as for the models described in [41] and [12]. We train on 16 TPU v2 chips.

For CLEVRER, we resize videos to 64 by 64 resolution and sample 25 random frames, as in [41]. We use two different MLP heads on top of the transformed value of the $CLS$ token to extract the final answer, one head for descriptive questions and one head for multiple choice questions. For descriptive questions, the MLP head outputs a categorical distribution over possible output tokens, whereas for multiple choice questions, the MLP outputs the probability that the choice is true. For

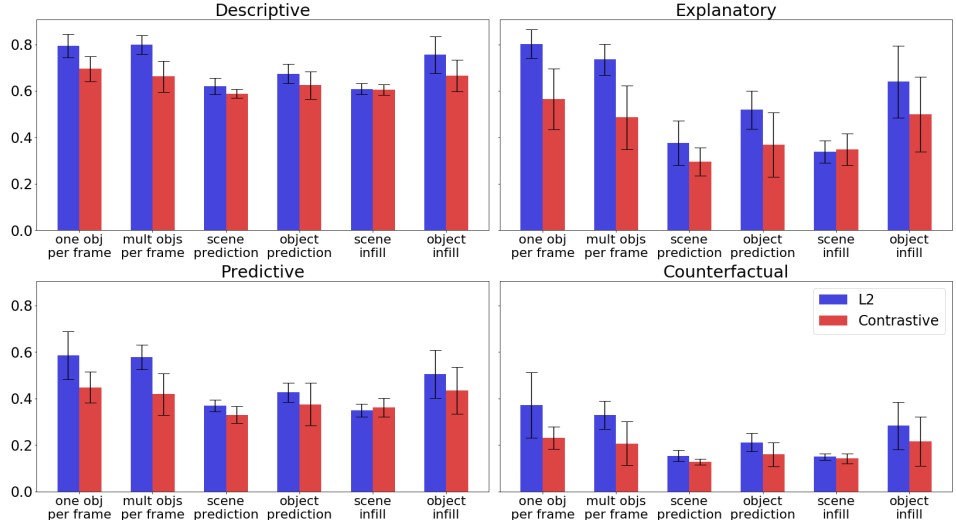

Figure 4: Comparison of different mask types and loss functions for auxiliary loss computation. Models were trained on 50% of the CLEVRER dataset to magnify the effects of the self-supervised loss.

each training step, we sample a supervised batch of 256 videos with their accompanying questions and answers along with an unsupervised batch of 256 videos, which do not include the answers. These batches are sampled independently from the dataset. The supervised batch is used to calculate the classification loss, and the unsupervised sub-batch is used to calculate the unsupervised auxiliary loss. This division was made so that we can use a subset of available data for the supervised batch while using all data for the unsupervised batch. The supervised batch is further subdivided into two sub-batches of size 128, for descriptive and multiple choice questions (this division was made since the output format is different for the two types of questions). *Aloe* converges within 200,000 training steps.

For CATER, we also resize videos to 64 by 64 resolution and sample 80 random frames. We use an MLP head on top of the transformed $CLS$ token. This head outputs a categorical distribution over the grid index of the final snitch location. We train on static and moving camera data simultaneously, with the batch of 256 videos divided equally between the two. *Aloe* converges within 50,000 training steps.

On ACRE, we resize each image to 64 by 64 resolution and concatenate the context images along with one query image to form a "video". The MLP head on top of the transformed $CLS$ token outputs a categorical distribution over the three possible answers: "yes", "no", and "undetermined". *Aloe* converges within 60,000 steps.

For the CLEVRER and CATER datasets, we pretrain a MONet model on frames extracted from the respective dataset. The training of the MONet models follow the procedures described in [2]. For ACRE, we reuse the MONet model we trained for CATER.

Motivated by findings from language modeling, we trained the main transformer model using the LAMB optimizer [42] and found that it offered a significant performance boost over the ADAM optimizer [22] for the CLEVRER dataset (data not shown). We use learning rate warmup over 4000 steps and a linear learning rate decay. We also used a weight decay of 0.01. All error bars are computed over at least 5 seeds. We swept over hyperparameters, and the below table lists the values used in our model. The hyperparameters we used for ACRE were the same as those we used for CATER, except that the prediction-head hidden layer size is reduced to 36 (from 144), because ACRE has only 3 possible outputs compared to the 36 for CATER. We did not do any hyperparameter tuning for ACRE.

| Parameter | Value | Parameter | Value |
|---|---|---|---|
| Batch-size | 512 | Batch-size | 256 |
| Transformer heads | 10 | Transformer heads | 8 |
| Transformer layers | 28 | Transformer layers | 16 |
| Embedding size $d$ | 16 | Embedding size $d$ | 36 |
| Number of objects $N_o$ | 8 | Number of objects $N_o$ | 8 |
| Prediction head hidden layer size | 128 | Prediction head hidden layer size | 144 |
| Maximum learning rate | 0.002 | Maximum learning rate | 0.002 |
| Learning rate warmup steps | 4000 | Learning rate warmup steps | 4000 |
| Final learning rate | $2 \times 10^{-7}$ | Final learning rate | $2 \times 10^{-7}$ |
| Learning rate decay steps | $2 \times 10^5$ | Learning rate decay steps | $5 \times 10^4$ |
| Weight decay rate | 0.01 | Weight decay rate | 0.01 |
| Infill cost $\lambda$ | 0.01 | Infill cost $\lambda$ | 2.0 |

(a) Hyperparameters for CLEVRER.      (b) Hyperparameters for CATER.

## B Using other object-segmentation algorithms

In the main text, we use MONet to obtain object representations, because MONet's unsupervised nature allows us to establish our state-of-the-art results using only data from the datasets. Our method of attention over learned object embeddings, however, does not rely on MONet representations in particular. In this section, we show how to apply our method to object detection models that output an object segmentation mask but not necessarily a feature vector for each object. This includes, for example, often-used models such as Mask R-CNN and DETR [3, 19].

Let $\mathbf{A}_{ti} \in [0, 1]^{w \times h}$ be the segmentation masks, either produced by an object segmentation algorithm or ground-truth masks. For any function $f : [0, 1]^{w \times h \times c} \to \mathbb{R}^d$ mapping from the image space to a latent space of dimension $d$, we can construct object feature vectors $\mathbf{v}_{ti} = f(\mathbf{A}_{ti} \cdot \text{image})$. That is, we apply $f$ to the image with the segmentation masks applied, once for each object. In our experiments, we choose to represent $f$ with a ResNet consisting of 3 blocks, with 2 convolutional layers per block. The weights of the ResNet are learned with the rest of the network, but the weights of the object segmentation model are fixed.

We provide a proof-of-concept using ground-truth segmentation masks to show the performance of our model in the ideal setting, independent of the quality of the segmentation model. We apply our model to the original CLEVR dataset [21], for which we have ground-truth segmentation masks. CLEVR is a widely used benchmark testing for understanding of spatial relationships between objects in a still image. We obtain an accuracy of **99.5%**, which is inline with state-of-the-art results (99.8%, [39]).

## C Analysis of CLEVRER dataset

During analysis of our results, we noticed that some counterfactual questions in the CLEVRER dataset can be solved without using counterfactual reasoning. In particular, about 47% of the counterfactual questions ask about the effect of removing an object that did not collide with any other object, hence having no effect on object dynamics; an example is given in Figure 5. Moreover, even for the questions where the removed object is causally connected to the other objects, about 45% can be answered perfectly by an algorithm answering the question as if it were a descriptive question. To quantify this, we wrote a symbolic executor that uses the provided ground-truth video annotations and parsed questions to determine causal connectivity and whether each choice happened in the non-counterfactual scenario.

Although determining whether or not a given counterfactual question can be answered this way still requires counterfactual reasoning, we want to eliminate the possibility that our model achieved its 75% accuracy on counterfactual questions without learning counterfactual reasoning; instead it might have reached that score simply by answering all counterfactual questions as descriptive questions. To verify this is not the case, we evaluated *Aloe* on only the harder category of counterfactual questions where the removed object does collide with other objects and which cannot be answered by a descriptive algorithm. We find that *Aloe* achieves a performance of 59.8% on this harder category.

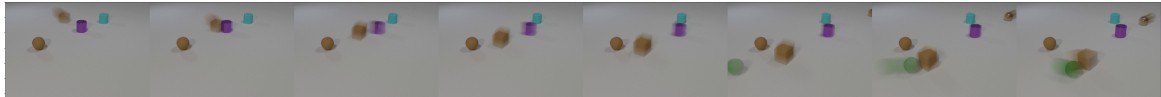

Figure 5: The video for an example counterfactual question that can be answered as if it were a descriptive question. The question is: if the brown rubber sphere is removed, what will not happen?

This is significantly above chance, suggesting that *Aloe* is indeed able to do some amount of true counterfactual reasoning.

# D    Qualitative analysis

We provide more qualitative analysis of attention weights in order to shed light on how *Aloe* arrives at its predictions. These examples illustrate broad patterns evident from informal observation of the model's attention weights. We focus on the following video from CLEVRER:

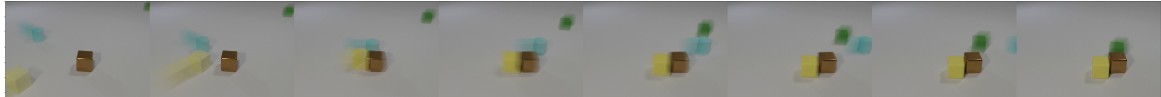

In this video, a yellow rubber cube collides with a cyan rubber cylinder. The yellow cube then collides with a brown metallic cube, while the cyan cylinder and a green rubber cube approach each other but do not collide. Finally, the green cube approaches but does not collide with the brown cube.

**Most important objects**    In the main text, we looked at the most heavily attended-upon objects in determining the answer to a counterfactual question about this video. By looking at the attention patterns when answering a different question about the same video (a predictive question, whether or not the cylinder and the green cube will collide), we see that the relative importance of the various objects depends on the question the model is answering. Here, we observe one head of the transformer focusing on collisions: first the collision of the cylinder and the yellow cube, then on the cylinder and the green cube when they move towards each other.

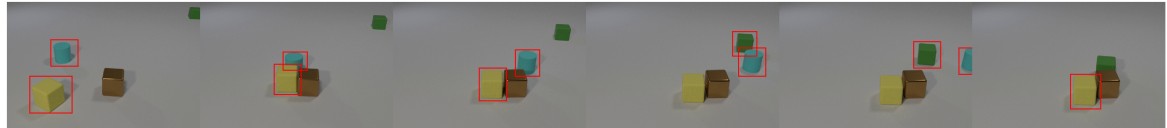

**Object alignment**    Recall that MONet does not assign objects to slots in a well-determined manner— tiny changes in an image can cause MONet to unpredictably assign objects to slots in a different permutation. This is a general flaw for object segmentation algorithms without built-in alignment. Nevertheless, *Aloe* can still effectively utilize these representations, because *Aloe* is able to maintain object identity even when the objects appear in different order in different frames. The image below, where we again show the two most attended-upon objects for each frame, illustrate instances where MONet changes the permutation of objects. In this image, we plot time on the x-axis and MONet slot index on the y-axis; the slots containing the two most important objects are grayed out. We observe that the transformer is able to align objects across time, maintaining consistent attention to the green and brown objects.

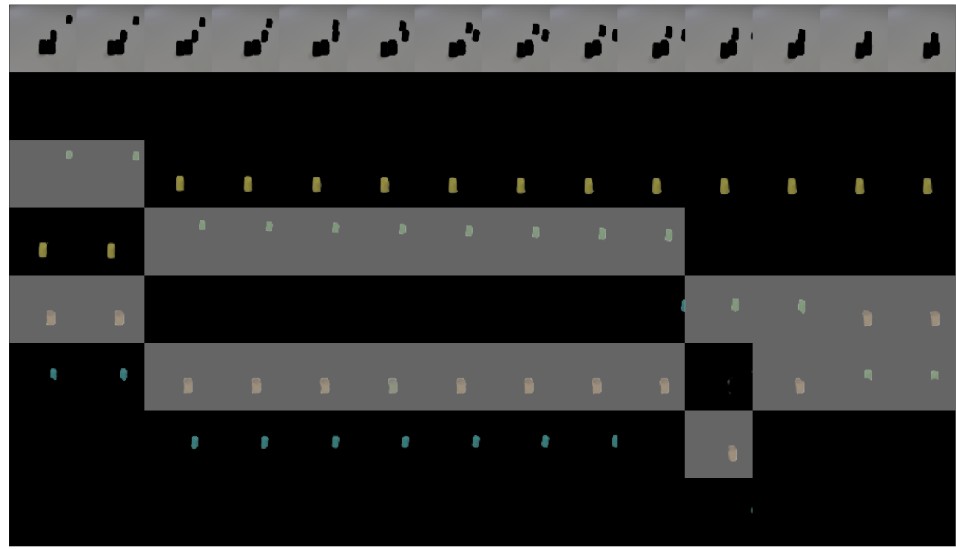

**Effectiveness of the auxiliary loss**  Finally, we visually inspect our hypothesis that our self-supervised loss encourages the transformer in learning better representations. For clarity of the subsequent illustration, we use the scene prediction masking scheme, as described in Figure 2. In this scheme, the transformer has to predict the contents of the last few frames (the *target frames*) given the beginning of the video. To pose harder predictive challenges, we mask out the three frames preceding the target frames in addition to the target frames themselves. The two images below compare the predicted frames (second image) to the true frames (first image). In the second image, the black frames are the three masked out frames preceding the target frames. The frames following the black frames are the target frames; they contain the MONet-reconstructed images obtained from latents predicted by the transformer. The frames preceding the black frames are MONet-reconstructed images obtained from the original latents (the latents input into the transformer).

We observe that with the self-supervised loss, we get coherent images from the transformer-predicted latents with all the right objects (in the absence of the auxiliary loss, the transformed latents generate incoherent rainbow blobs). We also observe the rudiments of prediction, as seen in the movement of the yellow object in the predicted image. Nevertheless, it is also clear that the transformer's predictions are not perfect, and we leave improvements of this predictive infilling to future work.

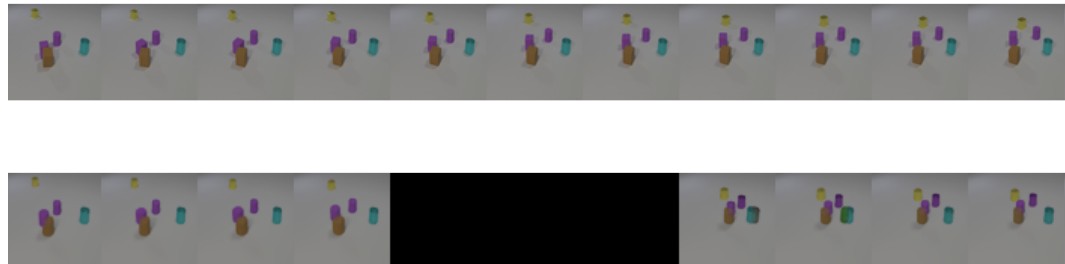

# E   Example model predictions

In this section, we provide a few sample classifications produced by *Aloe*. All examples are produced at random from the validation set; in particular we did not cherry-pick any examples to highlight the performance of *Aloe*.

We provide four videos and up to two questions per question type for the video (many videos in the dataset come with only one explanatory or predictive question). For each question type with more than one question, we try to choose one correct classification and one misclassification if available to provide for greater diversity. Besides this editorial choice, all classifications are sampled randomly.

**Q:** How many metal objects are moving?
**Model:** 1
**Label:** 1

**Q:** What is the shape of the stationary metal object when the red cube enters the scene?
**Model:** cylinder
**Label:** cylinder

**Q:** Which of the following is not responsible for the collision between the metal cube and the yellow cube?

1. the presence of the gray cube
2. the gray object's entrance
3. the presence of the red rubber cube
4. the collision between the gray cube and the metal cube

**Model:** 3
**Label:** 3

**Q:** Which event will happen next?

1. The gray object collides with the red object
2. The gray object and the cylinder collide

**Model:** 1
**Label:** 1

**Q:** Which event will happen if the red object is removed?

1. The gray object and the brown object collide
2. The gray object collides with the cylinder
3. The gray cube collides with the yellow object
4. The brown cube and the yellow object collide

**Model:** 1, 4
**Label:** 1, 4

**Q:** What will happen if the cylinder is removed?

1. The brown cube collides with the red cube
2. The red object and the yellow object collide
3. The gray cube collides with the red cube
4. The gray object collides with the brown object

**Model:** 3, 4
**Label:** 3, 4

**Q:** What color is the metal object that is stationary when the metal cube enters the scene?
**Model:** blue
**Label:** blue

**Q:** What material is the last object that enters the scene?
**Model:** metal
**Label:** rubber

**Q:** Which of the following is not responsible for the collision between the cyan object and the sphere?

1. the presence of the red rubber object

2. the red object's entering the scene

3. the collision between the sphere and the blue cube

**Model:** 1, 2, 3
**Label:** 1, 2, 3

**Q:** What will happen next?

1. The metal cube and the red cube collide

2. The sphere collides with the metal cube

**Model:** 1
**Label:** 1

**Q:** Without the red cube, which event will happen?

1. The sphere collides with the blue cube

2. The cyan object and the blue cube collide

**Model:** 1
**Label:** 1

**Q:** What will not happen without the sphere?

1. The cyan object collides with the red cube

2. The cyan object collides with the metal cube

3. The metal cube and the red cube collide

**Model:** 1, 3
**Label:** 3

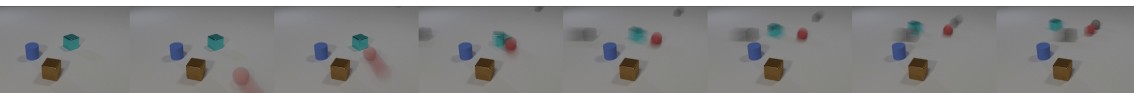

**Q:** Are there any moving brown objects when the red object enters the scene?
**Model:** no
**Label:** no

**Q:** How many rubber objects are moving?
**Model:** 3
**Label:** 3

**Q:** Which of the following is not responsible for the collision between the red object and the gray sphere?

1. the presence of the gray cube
2. the collision between the red object and the cyan object
3. the rubber cube's entering the scene
4. the presence of the cyan object

**Model:** 1, 3
**Label:** 1, 3

**Q:** What will happen next?

1. The gray cube and the brown object collide
2. The red object collides with the rubber cube

**Model:** 2
**Label:** 2

**Q:** If the cylinder is removed, which of the following will not happen?

1. The gray cube and the brown cube collide
2. The red object and the cyan object collide
3. The red sphere and the rubber cube collide
4. The cyan object and the brown cube collide

**Model:** 1, 4
**Label:** 1, 4

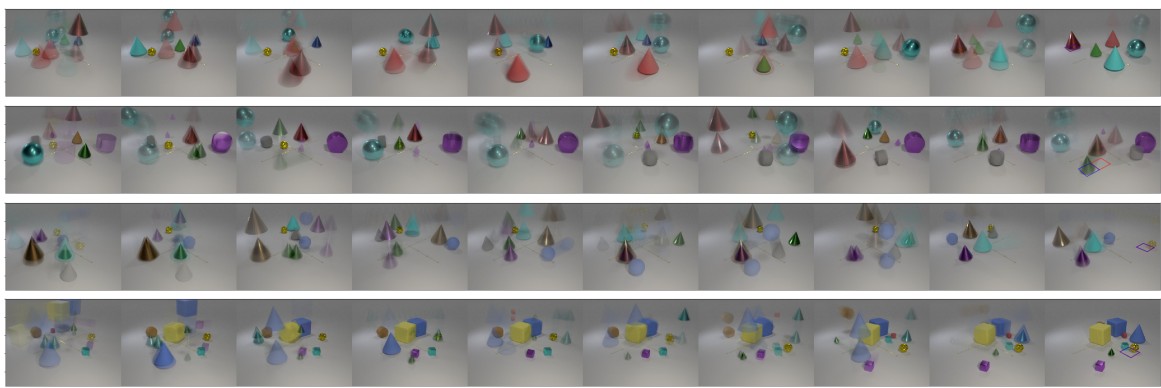

**Q:** How many objects are stationary when the sphere enters the scene?
**Model:** 1
**Label:** 1

**Q:** What is the shape of the last object that enters the scene?
**Model:** cube
**Label:** cube

**Q:** Which of the following is not responsible for the yellow object's colliding with the green object?

1. the presence of the purple sphere
2. the blue object's entrance
3. the collision between the blue object and the rubber cube
4. the sphere's entering the scene

**Model:** 2, 3
**Label:** 2, 3

**Q:** What will happen next?

1. The sphere collides with the rubber cube
2. The yellow cube and the green object collide

**Model:** 1
**Label:** 1

**Q:** Which event will not happen if the green cube is removed?

1. The yellow object and the blue object collide
2. The sphere collides with the blue cube
3. The sphere and the yellow object collide
4. The sphere collides with the yellow cube

**Model:** 2
**Label:** 2

**Q:** Which of the following will happen if the yellow object is removed?

1. The blue cube and the green cube collide
2. The sphere collides with the blue cube
3. The sphere collides with the green cube

**Model:** 1, 3
**Label:** 1

## E.2 CATER

We include ten random videos from the validation subset of the static camera CATER dataset. In the final frame of the video, the correct grid cell of the target snitch is drawn in blue, and the model's prediction is drawn in red. We note that the model is able to find the snitch in scenarios where the snitch is hidden under a cone that later moves (along with the still hidden snitch); in the sixth example, the model also handled a case where the snitch was hidden under two cones at some point in time.

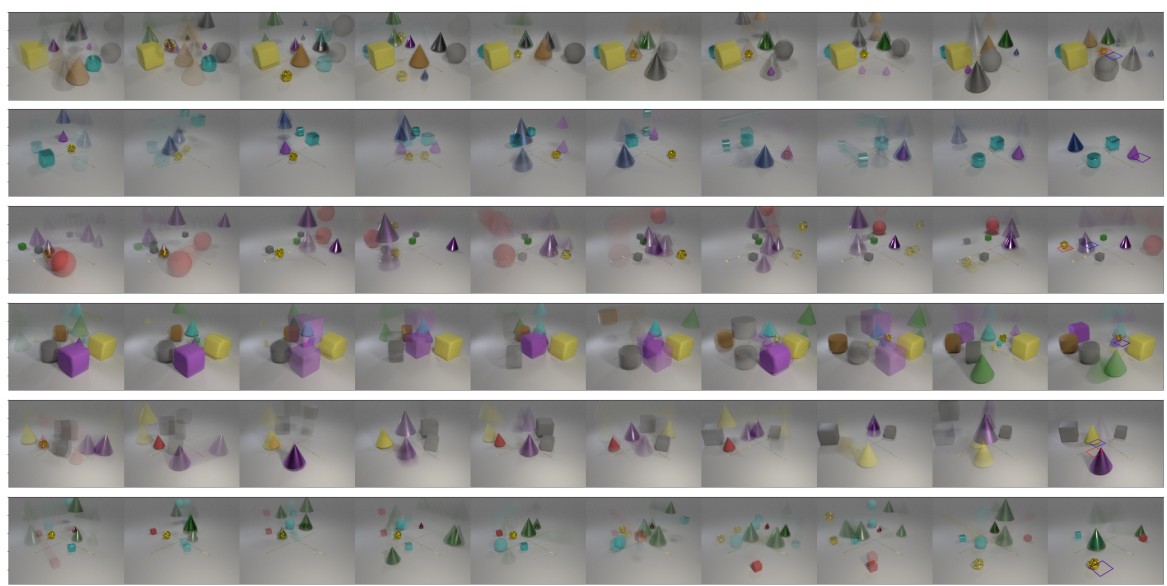

## F    Dataset Licenses

The CATER generation code is available under the Apache License, and the ACRE generation code is available under the GPL license.