# OpenReview forum: "Attention over Learned Object Embeddings Enables Complex Visual Reasoning"
_NeurIPS.cc/2021/Conference — NeurIPS 2021 Oral_

### Official Review · Reviewer_Un95 · 2021-07-13

**Rating:** 7
**Confidence:** 5

**Summary:**

The paper proposes to combine self-attention and object-centric representations to be able to solve complex reasoning tasks. The author(s) also propose a self-supervised loss based on masking of the objects' representations to implicitly learn the dynamics, further enhancing the performance on downstream tasks. The results of the proposed method are very impressive, especially on the challenging counterfactuals reasoning task.


**Main Review:**

Strengths:

1) The proposed model seems to be fairly simple with minimal inductive bias as compared to neuro-symbolic (NS) methods that assume the parser to be available. It is quite impressive that such a model can outperform NS based methods by a significant margin (especially on the harder causal reasoning tasks).
2) I'd also like to appreciate the author(s) efforts to share the model's code during the review process.

Questions:

3) Despite knowing the underlying dynamics explicitly, why do neuro-symbolic methods (like DCL/NS-DR) fail with such a large margin as compared to Aloe (~30% points)? Is this because of self-attention that the model can capture the long-term dependencies to reason, which MLP and even LSTM (from the footnote on Pg. 6) cannot?

4) Also, why is DCL/NS-DR not as sample efficient as Aloe given that DCL has enough (rather high) inductive bias of having a parser as well as ground-truth annotations for objects to learn from? For this case, we can exclude the counterfactual reasoning and only focus on the other three experiments. In Fig. 3, having DCL plots for different proportions for the dataset would be beneficial. (I don't think it is required for both CATER and CLEVERER datasets; just one should suffice)


5) One key aspect that NS methods claim is that of generalization to newer scenarios. For example authors in the DCL paper propose CLEVERER-grounding and CLEVERER-Retrieval tasks. I'm wondering how Aloe would generalize to such new tasks (need not be specifically these tasks).


Minor details:

6) Please include some details in the caption of Figure 1. Typically every figure should be stand-alone and should not require the reader to refer to the text.

7) Fig. 4 in the appendix shows results with 50% of CLEVERER data. Does the trend remain similar or does the gap reduce when trained with 100% of data (no need to show the results but I'm sure that an experiment with the complete data would've been run first)?

Justification for rating: Overall, the paper is well written and the proposed method though simple, is quite effective. My major concerns are with sample-efficiency and generalization compared to Neuro-Symbolic methods. However, I lean towards a weak accept of the paper as it does have some important claims regarding using a more general approach with less inductive bias for reasoning tasks.
My final rating however, would be based on the author(s) rebuttal as well as other reviewers' comments.

------
Edit (post-rebuttal) : All my concerns have been addressed satisfactorily, hence I lean towards the acceptance of the paper and am increasing my rating to 7 (accept). I thank the author(s) for their rebuttal.

**Time Spent Reviewing:**

9 hours

---

> ### Author Response · Authors · 2021-08-09
> **Response to Reviewer Un95**
>
> We thank you for your review and feedback. We address your questions below.
>
> 3. [*Why neuro-symbolic models underperform*] As neuro-symbolic models are not the focus of this work, we believe that providing a rigorous explanation for why they do not work as well is out of scope for this work, which is dedicated to showing a system that succeeds. We hypothesize that one reason the neuro-symbolic models underperform Aloe is that the symbolic component is rigid and not robust to errors in its inputs. The inputs to the symbolic executor are produced by several complicated networks (object detection, dynamics and collision prediction), which are all fallible. With Aloe, we use simpler "input networks" (object detection only as opposed to dynamics modeling) and the transformer is more robust to errors in object detection. We believe that this robustness is an essential feature for a reasoning model, as perfect perception and dynamics prediction is not realistically achievable.
> 4. [*Sample efficiency*] Our sample efficiency claim is more about the benefits brought by the self-supervised loss and not a claim that Aloe is necessarily more data-efficient than NS models. With self-supervision, Aloe requires only 60% of the supervised training data to match the performance of DCL, but we do not currently have data on how DCL scales with data. It is also not clear how we can compare supervised data usage between Aloe and DCL, as DCL also requires extra labeled data to train a language to symbolic program parser, whereas Aloe learns the semantics of the language input without this extra data. We will carefully review our text to make sure we do not imply that NS models are less sample-efficient than Aloe.
> 5. [*Generalizability*] We note that there are several types of generalization to consider:
>   * Generalization of the model architecture to new tasks/domains
>   * Zero-shot generalization to the same task but with different data distribution
>   * Zero-shot generalization to subtasks within the same domain
> CLEVRER, CATER, and ACRE are different task domains testing different skills and generated by different grammars. By contrast, the CLEVRER-Grounding and CLEVRER-Retrieval tasks are subtrees of the symbolic program defining the CLEVRER task.
>
> One strength of neuro-symbolic models is their modularity: a complex task like answering CLEVRER questions is broken down into simpler tasks that are easily isolatable, such as identifying objects satisfying a certain description (CLEVRER-Grounding). Because of this modularity, neuro-symbolic models can be adapted straightforwardly to all subtasks of the main task. In contrast, it is difficult to isolate and extract a subcomponent that does a specific sub-task in fully distributed models like Aloe, even if we suspect the model implicitly decomposes the task into subtasks (e.g. our attention weights suggest that our model can implicitly select for objects based on words or track objects across frames). This is indeed a weakness of connectionist models that we’ll be sure to include in our discussion section.
>
> As for the other two types of generalizability, we note that the Aloe architecture is capable of generalizing to a diverse set of tasks with re-training. Moreover, on ACRE, Aloe can correctly perform inference on scenes with a different distribution of causal factors (the systematic split of ACRE), suggesting that Aloe does not “take shortcuts” and overfit to do inference on only particular causal factors.
>
> 6. Thank you for your feedback, we will add more information to the caption to make the figure stand-alone in our final revision.
> 7. The trend remains similar, but the gaps are reduced. It is unsurprising that the effect is smaller. As seen in Figure 3, the maximum gap between Aloe with self-supervision and without to be at about 50% of the data. We would therefore expect the maximum gap between different variations of self-supervision to be greatest at that point too.

---

> > ### Comment · Reviewer_Un95 · 2021-08-23
> > **final decision from my end**
> >
> > I thank the author(s) for their rebuttal. All my concerns have been addressed in the rebuttal and after going over other reviews I've decided to increase my rating to 7 (Accept).

---

### Official Review · Reviewer_cuzz · 2021-07-17

**Rating:** 8
**Confidence:** 5

**Summary:**

This paper builds an end-to-end system for visual reasoning. The videos are first parsed to object-centric representations using an unsupervised object detection method (MONet). After that, they use a transformer on top of the object-centric representations to model the spatio-temporal relationship. The features are then combined with the question embedding and output the answer. The learning process is driven by two losses: 1) self-supervised prediction (the authors examine different forms of such supervision); 2) a supervised-learning loss such as the answer to questions. The proposed method achieves impressive improvement over previous state-of-the-art.

**Limitations And Societal Impact:**

I didn't find any potential negative societal impact of this work.

**Main Review:**

The authors propose a simple framework that can achieve impressive performance on many visual reasoning benchmarks. The proposed method holds the potential to be a simple baseline for future research in this direction because of its simplicity, elegance, and good performance. The model use transformer to simultaneously process object information and language information, which leads to significant performance improvement and makes a lot of sense.

Indeed, as the authors mentioned in the submission history, there exists criticism about the visual reasoning benchmark itself (such as CLEVERER). However, the contribution and originality of this work should not be omitted. One thing I want to mention is that the current leading neural-symbolic methods in this dataset (such as [5] and [37]) usually imposes human priors specifically for this simple environment (such as it only contains blocks and balls). However, an important point is that the proposed method does not have any human priors thus it holds the potential to be applied to real-world. Maybe the only component that cannot easily generalize to real-world scenario is the MONet which people find it hard to apply to real-world videos. However, with the advanced supervised object detection techniques, it’s still highly possible towards that goal, as also shown in [A].

In conclusion, I think it’s a good paper in this field and can potentially help the neural-symbolic methods to get new improvements.

[A] Learning Long-term Visual Dynamics with Region Proposal Interaction Networks. ICLR 2021


**Time Spent Reviewing:**

5.0 hours

---

> ### Author Response · Authors · 2021-08-09
> **Response to Reviewer cuzz**
>
> We thank you for your positive review. We are glad that you appreciate the significance of our work. Please let us know if you have any suggestions on how we can improve our paper.

---

### Official Review · Reviewer_WrDi · 2021-07-23

**Rating:** 9
**Confidence:** 5

**Summary:**

A transformer achieves state of the art in three video+text reasoning tasks. A pre-trained unsupervised method obtains object representations per frame; object representations across frames, word representations (question+options) and a special token is fed to a transformer and a prediction is made using a linear layer on top of the transformed special token. Question answers provide supervision and masking objects provide self-supervision. Ablation experiments are performed: they show the usefulness of object-centric representations and, albeit less so, self-supervision. Model attention and errors seem sensible. It shows that a fully end-to-end network can solve causal reasoning tasks; which until now required symbolic modules such as program generation and execution engines or physical engines.

**Limitations And Societal Impact:**

Good job pointing the limitation of using only synthetic datasets. I would go even further saying it has only being applied to the same syntethic environment (object identities, colors, shapes, size, material, background, illumination, etc.): all CLEVR-like. Can you venture a prediction, perhaps from experience, about whether it will transfer to other synthetic or real environments?

The one-sentence ethical statement seems haphazard; picking data biases is surely a pitfall of any learning method but methods like the one proposed here that eases observation/tracking of activity in video have more present concerns: facilitating population tracking, for instance, or (through better causal modelling) artificial manipulation of public opinion. Training big models, with the usual disregard for the amount or source of energy used, could also be brought up.

**Main Review:**

Originality
Models (unsupervised object representation, self-supervision, transformers) and datasets are current; the novelty of the paper is in combining them to show that “simple” architectures can perform visual reasoning/causal tasks without  explicit need for a symbolic reasoning engine.

Quality
Good paper with well-thought, thorough experiments and positive results.

Ablation experiments, error bounds, and qualitative evaluations provide assurance on the validity of results and claims made. Provided hyperparameters, training details and code will help replicability.

By design, you allow objects to attend to words and (in multi-option questions) words from one option to attend to words from other options which seems counterintuitive, did you ever try to use an encoder-decoder architecture (word-to-word, object-to-object and word-to-object attention), sending the questions as query in the decoder (or perhaps question+each_option as a query)? This is also a standard transformer setup and the extra structure may benefit the model.

Managing the information flow during masking becomes troublesome when you have rolling (frame-dependent) masks (c-f in Fig. 2). For instance in scheme c, even though in the first transformer layer representation slot i cannot access the masked slots at i, i-1 and i -2, in the second layer it will be able to access all representations, most of which have access to i, i-1 and i-2 thus bypassing the masking (I don’t see how one can mask the second layer to avoid this leaking of information.). This should not be a problem for a, b; maybe avoiding the allocation of informational capacity to this shortcutting is why a, b masking performs better.

In Tab. 1, the comparison of object-centric representations with representations from a ResNet trained from scratch is unfair; a ResNet-like architecture pre-trained with self-supervision (e.g., a contrastive loss) in the CLEVRER data would be better, or even a standard VAE encoder (perhaps with some disentanglement regularization to encourage object separation). In any case, using an object-centric representation seems more principled.

Although the paper advances the point of connectionist models being enough to solve reasoning tasks, in fairness, you could still point to some of the advantages that symbolic and neuro-symbolic models have over purely learned models: modularity, interpretability, human-computer interaction, etc. or contrast them more in the discussion.

Clarity
Well-written, readable and consistent figures.

First paragraphs in 3.2 and 3.3 describing the datasets are hard to understand without already being familiar with the datasets (which the paper relies on). I don’t think the authors can do much more as the responsibility falls on the readers but it does still take away from the manuscript. For instance, it is hard to contextualize what the results of Table 3 mean without a very good understanding of ACRE; observation holds for the other datasets too, though they are better known. Whether it is worth to present an example (or just pointing to the appendix examples) or to guide readers through the significance of results is up to the authors. I, of course, understand finding a good balance is hard.

Good explanation of the model architecture (Sec. 2). References to the appendix for extra details are welcomed.

For object collision and tracking datasets, object position (inside the frame) is very important. Perhaps, it is worth explicitly mentioning how this xy position is provided to the transformer; as I understand, in the distributed Monet representation (and not, for instance, provided in two separate dimensions or using the mask somehow). Regarding Monet, it is also unclear how you select the number of object slots and whether you discard the extra “empty” slots (and how).

Fig. 1 conveys a lot of information about the model architecture. I can point a number of small details (none too important): 1) arrows and SSL schematic are hard to see in print, perhaps you can decrease the transparency (or pick a different color), 2) you say you use a MLP on top of the CLS class for predictions (l. 120) but the figure says linear+softmax, 3) If model details are not in the caption, perhaps you could point directly to “see Methods/section 2 for details” rather than “see the main text” and 4) I wonder how important is it to have the “+position encoding” in the main transformer block, having it so front and center makes it look more important than it is and other details (such as the extra dimensions pointing whether the representation is a word or object) are not in the diagram either.

It is odd that half a page and a figure is devoted to explaining the different masking strategies and the results are not presented in the main text. Effect of the supervised loss is minor (as shown by ablation experiments) and it does not seem like the effect of different masking is that significant anyway (Fig. 4 in Appendix); maybe you just present the one-object per-frame masking in the main, bump the rest to appendix and more results from CLEVR (like the counterfactual analyses) can be upgraded to main. I do not feel strongly about this, though, it just feels discordant to spend a lot of space describing the method in main to then not show the results.

How do you do the object-to-slot matching when masking more than one object per frame? is it in the appendix?

Missing info: What are the error bounds in Tab. 1, Tab.2 and Fig. 3? stddev, stderr, ci? and how many samples are used?

Figure 3 could have bigger font sizes and thicker lines; it is relatively hard to read. The light/dark yellow bars are also hard to see (and one dissapears behind the other sometimes); perhaps using two side-by-side bars would help (or have them as horizontal lines in the plot). Also, consider adding a y axis at the very right of the plot (or turn grids on) as it is hard to see, for instance, how high the accuracy on counterfactuals goes in the last column.

It is unclear whether the Aloe results in Tab 1 (middle row) use infill or one object per frame masking? You say “infill” improves 4-6% (l. 217) but then say one object per frame is the most effective (l.222).

Which transformer layer is used for the attention visualizations in l.189?

The non-captioned figures in l.193 and l.200 are visually off-putting; I guess you were constrained on space; putting them together in the same figure could be as economical and underline the fact that the two images are a set (i.e., the question in image 1 is answered given the video in image 2).

For the CATER dataset, it is not fully clear what is the supervision signal for the transformer (from the original paper I surmised it is to predict the executed actions, is this correct?).  From the text, it seems like the supervision would be the distance between predicted and target snitch position (no mention of questions), which makes the addition of an L1 loss confusing (as it will be redundant).

Typo (l.285): representations

Nice literature review. I am curious to know why is it near the bottom and not between intro and methods where it would usually be. The self-supervised learning section is somehow light (for instance, word2vec used infilling long before BERT and other SSL methods have been used before contrastive learning), but it may be enough.

(Appendix) May be worth stating that code is written in Tensorflow and whether implementation is TPU specific? It will be nice to state which hyperparameters were searched, some more details about word embeddings (pretrained, vocabulary treatment, etc.) and how the positional encoding was done (I presume all slots in one frame have the same encoding, and then words start encoding at the start of the sinusoidal series).

Significance
Paper with good results that allow it to make bold claims: that end-to-end networks can model reasoning and causal tasks (at least the ones tested) without need for symbolic modules. It will be a good reference/baseline for future work in this subfield that is growing in popularity. Its applicability to realistic images as noted in the manuscript is left unadressed. Addressing some minor details will make the paper stronger but in its current state, acceptance as poster is still merited.


Update:
Bumped the rating from 7 to 9 (top 15% of papers) after the authors improved the manuscript using reviewers' input.

**Time Spent Reviewing:**

9

---

> ### Author Response · Authors · 2021-08-09
> **Response to Reviewer WrDi**
>
> We thank you for your detailed review and feedback. Your suggestions on improving the clarity of the text and our figures are very much appreciated, and we will incorporate them into our final camera-ready version. We will also add a more nuanced discussion of limitations, comparison with symbolic models, and ethical implications.
>
> We agree that the encoder-decoder setup you suggested would be an interesting alternative architecture to try. Just a minor clarification: in our setup, words from different choices do not attend to each other. We apply the transformer independently to each (question, choice) pair, for all the choices for that question. Essentially, we treat each choice of a multiple choice question as an independent question and ask the transformer to output if that particular choice is true or false. For CLEVRER, this accurately reflects that each option is true or false independently of the other ones.
>
> To clarify our masking scheme, we do not have rolling frame masking schemes. We mask elements of the input sequence to the transformer and not the attention weights, and so there is no leakage from masked out frames when going up a layer. The masked out frames in schemes c-f are chosen absolutely and are not relative to the current frame. Regarding how we match objects and slots when more than one slot is masked, this is something that we do not do at the moment which could improve the performance of masking schemes d and f. We leave this exploration for future work.
>
> *Re: fairness of the ResNet ablation.* We agree that pretraining the ResNet would be an interesting comparison. We weren’t sure what type of contrastive loss you were suggesting, but we will try pretraining the ResNet on ImageNet classification or via auto-encoding as you suggested.
>
> *Re: generalization to non-CLEVR synthetic images.* In our ongoing research we have had experience applying Aloe to various different environments and observe that it works well whenever object segmentation works well.
>
> *Re: how xy information is encoded in the object slot.* As you said, this is all done implicitly in the MONet object representations. The number of object slots is chosen to be an upper bound of the total number of objects; in practice we chose 8 for all three tasks. We do not explicitly discard empty slots, but cursory examination of attention weights suggests that these slots are treated differently from occupied slots.
>
> *Re: supervision signal for CATER.* In CATER, objects are located in an x-y-z coordinate system, where x and y range from -3 to 3. The xy plane is divided into a 6 by 6 grid, and the dataset provides the grid index of the snitch in the final frame. We predict this grid index and have both a classification loss (cross entropy over the 36 possible grid indices) and an L1 loss (L1 distance between predicted grid cell and the true grid cell). We acknowledge that this is not clear from our description of CATER. We will provide more details about CATER and the other datasets (possibly in the Appendix as you suggested).
>
> Our error bounds are all standard deviations, computed from at least five seeds for each error bar.
>
> Our attention visualization uses the last layer of attention weights.
>
> *It is unclear whether the Aloe results in Tab 1 (middle row) use infill or one object per frame masking?*
>
> We use one object per frame. Line 217 uses infill in a different sense to how we use it in Section 2.1 and Figure 2 (what we meant to say is, the self-supervised loss of infilling missing object slot, not the specific infilling objective of schemes e and f). We agree that this is very confusing, and we will drop the parenthetical “infill”.

---

> > ### Comment · Reviewer_WrDi · 2021-09-03
> > **answer to authors**
> >
> > Thank you for your response (I am aware it was a lengthy review).
> >
> > Re CLEVRER: The q+option method seems sensible. I was confused by l.546-547, I assumed the only difference between MLP heads for descriptive vs other questions was the softmax (categorical) vs logistic activation (set of bernoullis). Perhaps you can add the detail about each option being processed separately here.
> >
> > re masking: If I understand correctly, for schemas c-f only one frame of the entire video has self-supervision (i.e. not "relative to the frame") unlike a-b where the loss is applied at each frame. This is still rather unclear to me from the text (l.152) and to be frank, I still believe this section does not add much to the paper. In the interest of time (my bad) and given that the best masking scheme is single object per frame which is understandable and straightforward, I leave it to the authors to deal with this as they see fit.
> >
> > re ResNet pretraining: It is important the pretraining is done with CLEVR-like data so it learns to represent CLEVR-like objects (which an Imagenet ResNet might not be able to). I did not offer specific experiments for fear of being seeing as too "interfery". Contrastive tasks using CLEVRER could use, for instance, two frames from the same video as positive examples vs a frame from another video but, perhaps, self-supervised tasks could be better such as the ones you use masking object/scenes, or predicting next frames or number of objects in an scene or class of the objects in the scene, etc. You could also use the ImageNet-pretrained ResNet as a starting point (to then be finetuned given one of the tasks above). Any task should suffice, as long as you can say that the network has seen similar data to the data seeing by MONEt, you should be safe from criticism.
> >
> > re: generalizability: It is ok if you add "preliminary experiments show Aloe translates to other environments" or something along those lines. Up to the authors.
> >
> > re: xy information: What I was recommending is to make sure at any point in the paper to say that the object representation also carries information about the object location (as unfamiliar readers might think it only carries "object" information such as shape, color, size, etc.). re slots: sounds good (it seems customary to just leave the empty spaces unbothered).
> >
> > re: CATER: thank you very much for the explanation; i was not aware that the grid prediction was done as a classification problem.
> >
> > re: error bounds: Is this info in the paper? it should. (Same with the layer used for the attention visualization although less important).
> >
> > Overall, I think this changes will make an already good manuscript better!

---

> > > ### Author Response · Authors · 2021-09-06
> > > **Thank you for your review**
> > >
> > > Thanks again for your detailed comments – your review is incredibly helpful in helping us improve the quality of our paper. We will go through your comments carefully when revising our paper. Also, thank you for clarifying your point about additional pretraining. We weren't sure if you were looking for anything specific, but we now understand the general spirit of your suggestion and will carry out the experiment.

---

### Official Review · Reviewer_1FAX · 2021-07-28

**Rating:** 8
**Confidence:** 4

**Summary:**

The authors have proposed a model for spatiotemporal reasoning problems using self-attention and self-supervised learning in this work. For object representation, an unsupervised method, MONet (Burgess et al., 2019) is used where recurrent attention mechanisms are used to obtain N attention masks. This self-supervision along with unsupervised auxiliary loss helps in learning better representations. Experiments were performed with datasets like CLEVERER, CATER and ACRE after pretraining the model on individual frames. Through various experiments, authors have demonstrated the importance of self-attention, object-based discretization and auxiliary loss in their model. Overall, the model can perform higher-level reasoning with its simplified architecture.


**Limitations And Societal Impact:**

I feel that the model has certain limitations like:
1. Prior knowledge about the number of objects (N0) is required.
2. Pre-training is required for MONet on each dataset.

I am unable to understand how the authors have set the hyperparameters like transformer heads/layers, embedding size etc for their models. I am also not sure about the statement made in line 186; Was there any study showing that humans and models would have given incorrect responses for questions? Can you explain global vs hierarchical attention which is not very much clear from the paper? Finally, do you have a comparison of the total number of trainable parameters in ALOE vs other baseline models which might also affect the performance.

Minor:
https://arxiv.org/pdf/2006.15055.pdf [slot attention] reference is missing which I believe has a similar approach of finding the object-centric representation in an unsupervised manner.

**Main Review:**

The paper is clearly written and easy to follow. Wide ranges of experiments were performed to cover various aspects of reasoning using different datasets and have clearly shown an edge over previous SOTA models in a data-efficient way. Their ablation studies show the importance of different modules in their model.



**Time Spent Reviewing:**

10

---

> ### Author Response · Authors · 2021-08-09
> **Response to Reviewer 1FAX**
>
> We thank you for your review and feedback. The Slot Attention paper is definitely very relevant and will be included in our related works section.
>
> 1. [*Prior knowledge about the number of objects (N0) is required*]
> This is indeed a limitation of our model, though we note that we can choose N_O to be larger than the total number of objects — Aloe can handle empty slots. Indeed, for all tasks we choose N_O=8, even though the tasks have different numbers of objects. Having to specify an upper-bound on the number of objects/slots is a common drawback for object/slot models, including (for example) slot attention models.
>
> 2. [*Pre-training is required for MONet on each dataset*]
> Although we found pre-training MONet on each dataset separately helps slightly with the final performance, it is not required for Aloe to perform reasonably — indeed, we did not train a new MONet for ACRE. We previously tried using a single MONet model that was pre-trained on CLEVR, and we got results that are about (depending on the task) 5% worse than the final results with task-specific pretraining. We believe that this difference is caused by subtle visual differences between the tasks. For example, objects in CLEVR tend to be located further apart from each other than in CLEVRER (which feature many collisions), and the key CATER object, the snitch, is smaller than typical objects in CLEVR.
>
> We did not do a formal study with human raters on the qualitative performance of Aloe. Line 186 represents our own anecdotal experience with our model. To allow readers to form their own opinions, we provide a few examples in Appendix E of Aloe answering CLEVRER questions (all sampled at random from the dataset).
>
> Global and hierarchical attention refer to how objects in different frames attend to each other. In hierarchical attention, we have two transformers. The first operates on each frame independently, transforming objects representations of a single frame. The second transformer operates on the frame representations. By contrast, in global attention, a single transformer acts on the flattened array of objects across frames. In Python pseudo-code:
> ```
> # Let objects be a [batch (B), num frames (F), num objects (N), latents per object (D)] tensor.
> # Hierarchical
> outputs = transformer1(tf.reshape(objects, [B * F, N, D]))
> outputs = transformer2(tf.reshape(outputs, [B, F, N * D]))
>
> # Global
> outputs = transformer(tf.reshape(objects, [B, F * N, D]))
>
> ```
> We recognize that our current phrasing is potentially confusing, and we will rephrase our description in the paper to make it clearer.
>
> Regarding hyperparameter choices, we did a sweep over various hyperparameter values and chose the values that led to highest validation set performance. For ACRE, we used the same settings as CATER, as these hyperparameters already led to performance significantly higher than previous SOTA. We described this procedure briefly in Appendix A, but will clarify our description and refer to it from the main text.
>
> Finally, regarding the number of parameters, Aloe uses 15M trainable parameters for CLEVRER, 16M for CATER, and 16M for ACRE. Of the baseline methods across all our tasks, only Hopper [1] reports the number of parameters, which is 6.4M for CATER. While ALOE does use more parameters, we believe that this is offset by its flexibility, superior performance, and conceptual simplicity. Because ALOE does not have inductive biases for object permanence built-in, it will naturally require more parameters but can be more easily adapted to other tasks.
>
> [1] Honglu Zhou, Asim Kadav, Farley Lai, Alexandru Niculescu-Mizil, Martin Renqiang Min, Mubbasir Kapadia, and Hans Peter Graf. Hopper: Multi-hop transformer for spatio-temporal reasoning. ICLR, 2021. https://arxiv.org/abs/2103.10574

---

> > ### Comment · Reviewer_1FAX · 2021-08-25
> > **Final decision**
> >
> > I thank the authors for their explanation. Based on all the reviews and the author's response I have updated my ratings to 8.

---

### Decision · Program_Chairs · 2021-09-27

**Decision:**

Accept (Oral)

**Comment:**

This paper received 4 strong accepts. The reviewers have lauded the work because of extensive experiments and clear cutting results which are pushing the SOTA. The approach is likely to be of broad interest because this "connectionist" approach is pitted again the "symbolic" one in an area where so-called neuro-symbolic models have been dominating. The AC thus recommends an oral.